# Evidence of Influenza A Virus Infection in Cynomolgus Macaques, Thailand

**DOI:** 10.3390/vetsci9030132

**Published:** 2022-03-13

**Authors:** Weena Paungpin, Metawee Thongdee, Natthaphat Ketchim, Somjit Chaiwattanarungruengpaisan, Aeknarin Saechin, Ladawan Sariya, Supakarn Kaewchot, Pilaipan Puthavathana, Witthawat Wiriyarat

**Affiliations:** 1The Monitoring and Surveillance Center for Zoonotic Diseases in Wildlife and Exotic Animals, Faculty of Veterinary Science, Mahidol University, Nakhon Pathom 73170, Thailand; weena.pau@mahidol.edu (W.P.); metawee.tho@mahidol.edu (M.T.); charinrat.ket@mahidol.ac.th (N.K.); somjit.cha@mahidol.ac.th (S.C.); aeknarin.sae@mahidol.ac.th (A.S.); ladawan.sar@mahidol.edu (L.S.); 2Department of National Parks, Wildlife and Plant Conservation, Bangkok 10900, Thailand; supakarn_vet@hotmail.com; 3Center for Research and Innovation, Faculty of Medical Technology, Mahidol University, Nakhon Pathom 73170, Thailand; pilaipan.put@mahidol.ac.th

**Keywords:** nonhuman primates, cynomolgus macaques, influenza A virus, surveillance, Thailand

## Abstract

Little is known about the ecology of influenza A virus (IAV) in nonhuman primates (NHPs). We conducted active surveillance of IAV among 672 cynomolgus macaques (*Macaca fascicularis*) living in 27 free-ranging colonies in Thailand between March and November 2019. A hemagglutination inhibition (HI) assay was employed as the screening test against 16 subtypes of avian influenza virus (AIV) and two strains of the H1 subtype of human influenza virus. The serum samples with HI titers ≥20 were further confirmed by microneutralization (MN) assay. Real-time RT-PCR assay was performed to detect the conserved region of the influenza matrix (*M*) gene. The seropositive rate for subtypes of IAV, including AIV H1 (1.6%, 11/672), AIV H2 (15.2%, 102/672), AIV H3 (0.3%, 2/672), AIV H9 (3.4%, 23/672), and human H1 (NP-045) (0.9%, 6/672), was demonstrated. We also found antibody against more than one subtype of IAV in 15 out of 128 positive tested sera (11.7%). Moreover, influenza genome could be detected in 1 out of 245 pool swab samples (0.41%). Evidence of IAV infection presented here emphasizes the role of NHPs in the ecology of the virus. Our findings highlight the need to further conduct a continuous active surveillance program in NHP populations.

## 1. Introduction

The influenza A virus (IAV) is a paradigm for an emerging infectious virus that is still evolving. The gene pool of IAV in aquatic birds provides all the genetic diversity required for the emergence of the epidemic/pandemic influenza virus in reservoir hosts [1]. Molecular characterization and epidemiological study suggests that the pandemic human influenza viruses originated mostly from animals, in particular, pigs and birds [2]. The phenomenon has been previously observed in several past human influenza pandemics. The continuous circulation in animal hosts could make the influenza viruses undergo periodic genetic changes (evolution, adaptation, and gene reassortment), leading to the possibility of the emergence of new strains with epidemic and pandemic potential [3]. Thus, surveillance of IAV infection in many animal species, especially the animals that live closely with humans, is still needed.

Nonhuman primates (NHPs) are animals that have physiological and immunological similarities with humans. As such, NHPs have been used for decades as animal models to study immune responses, viral vaccine efficacy, and antiviral drugs against avian and human influenza viruses [4,5,6,7,8,9]. However, little is known about IAV infection in NHP populations in nature. Previous studies have demonstrated the infection of IAV different subtypes among NHP populations. Seropositivity to avian influenza virus (AIV) subtype H9N2 virus and seasonal human influenza subtypes, H1N1 and H3N2 viruses, have been detected in macaques in Bangladesh, Singapore, and Indonesia [10]. Antibodies to H1N1 and H3N2 have been evidenced in captive chimpanzees in the Netherlands [11]. Moreover, the influenza virus genome has been found in a buccal swab from an adult macaque in Cambodia [10]. In contrast to other studies, our previous study found no antibodies against AIV subtypes H1N1, H3N8, H5N1, and H7N1, and the seasonal human H1N1 virus in serum samples derived from 109 macaques collected during 2009–2017 in Thailand [12]. Moreover, the result of real-time RT-PCR demonstrated no influenza A viral RNA detected in the 282 NHP oropharyngeal swab samples collected in 2018 [12]. 

In Thailand, cynomolgus macaques (*Macaca fascicularis*) are the most frequently observed species among the 13 species of primates [13]. In 2018, the Department of National Parks, Wildlife and Plant Conservation reported the presence of about 50,000 individuals in Thailand living in the area with human–macaque conflict. In this case, cynomolgus macaques were the main population accounting for 92.25% of the total population [14]. Many free-ranging macaque colonies live closely or share the same habitats/environment with domestic animals, wild animals, and human communities. The close proximity of macaque colonies and the human community could pose a potential risk for virus transmission between macaques and humans. A surveillance program is important for monitoring a potential zoonotic threat such as influenza viruses. Thus, we performed an investigation on IAV exposure using serological and molecular methods in free-ranging cynomolgus macaques in Thailand between March and November 2019. 

## 2. Materials and Methods

### 2.1. Ethical Approval 

The use of samples for this study was approved by the Faculty of Veterinary Science, Mahidol University Institute of Animal Care and Use Committee (FVS-MU-IACUC); animal Ethics No. MUVS-2019-12-55. 

### 2.2. The Study Population and Sample Collection

The sample collection of macaques was carried out in the field site between March and November 2019 under the cooperation of the Department of National Parks, Wildlife and Plant Conservation. The sampled animals were living in free-ranging colonies distributed throughout Thailand. Most of the animals showed no respiratory diseases or clinical symptoms at the time of sample collection. Demographic data on the sex and age of individual animals were recorded (Table 1). After general anesthesia, animals were subjected to blood and swab collection. Whole blood was drawn via saphenous/cephalic vein into plain tubes and allowed to clot at ambient temperature for at least 30 min before storing at 4 °C. Buccal, oral, oropharynx, or rectal swabs were placed into individual tubes of 1 mL phosphate buffer saline (PBS). All samples were kept at 4 °C and transported to the laboratory within 48 h. The separation of serum from the clotted blood was performed after the sample arrived in the laboratory. The serum samples were subsequently kept at −20 °C until analysis. 

Overall, the specimens were taken from a total of 672 macaques located in 27 free-ranging colonies in 16 provinces. Generally, 25 animals were sampled per colony. Most of the macaque colonies were found in close proximity to human communities such as villages, primary schools, temples, and tourist places. Of those 672 sampled macaques, approximately 35% (233/672) and 53% (355/672) of the animals were sub-adult and adult, respectively while 12% (84/672) of the animals lacked age data. More than 60% (422/672) of the animals were male and nearly 30% (192/672) were female. No available sex data were found in 9% (58/672) of the animals. 

### 2.3. The Study Viruses

Sixteen subtypes of the AIV virus and two strains of the human H1N1 influenza virus were employed for HI and MN tests. The list of tested viruses in this study is shown in Table 2. To prepare the virus stocks for use in the HI and MN assays, all AIV viruses were propagated in embryonated eggs, while human H1N1 influenza viruses were propagated in Madin–Darby canine kidney (MDCK) cells (obtained from the American Type Culture Collection; CCL-34) maintained in the viral growth media containing Eagle’s minimum essential medium (EMEM) (Gibco, Grand Island, NY, USA) supplemented with 2 μg/mL of trypsin–tosyl phenylalanyl chloromethyl ketone (trypsin-TPCK) (Sigma–Aldrich, St. Louis, MO, USA).

### 2.4. Hemagglutination Inhibition (HI) Assay 

A hemagglutination inhibition (HI) assay was performed as a screening test for the detection of antibodies against 16 subtypes of the AIV virus, including H1-H16 virus and two strains of human H1N1 influenza virus comprising human H1 (104) virus and human H1 (NP-045) virus. All serum samples were treated with receptor destroying enzyme (RDE; Denka Seiken, Japan) at 37 °C for 16–18 h, followed by heat inactivation at 56 °C for 30 min and absorbing with 50% goose erythrocyte suspension at 4 °C for 1 h. The HI assay was carried out by the procedure described previously [15,16]. The test was performed twice and duplicate wells were run for screening dilution at 1:10 of individual serum samples. The treated serum control, red blood cells (RBCs) control, and back titration of the test virus were included in each run. Geometric mean titers (GMTs) of each virus subtype were calculated. The HI titers *<* 20 were assigned as 10, and the HI titers ≥ 320 were assigned as 320. The serum samples with HI titers ≥ 20 were further determined for neutralizing antibodies using a microneutralization (MN) assay. 

### 2.5. Microneutralization (MN) Assay 

The MN assay was conducted as previously described [16,17]. Briefly, the treated serum was mixed with the test virus (final concentration of 100 TCID50/well) and incubated at 37 °C for 2 h. The serum–virus mixture was transferred onto the MDCK monolayer cells and then further incubated at 37 °C for 2 d. The cell monolayers were examined for the appearance of cytopathic effect (CPE), while the culture supernatants were determined for non-neutralized viruses by a hemagglutination assay. The neutralization titers (NT) were determined as the reciprocal of the last dilution neutralizing the virus replication. The serum samples showing both of the HI and NT titers ≥20 were considered positive.

### 2.6. Real-Time Reverse Transcription-PCR

Viral RNA was extracted from the pooled swabs using a QIAamp Viral RNA Mini Kit (QIAGEN Inc, Valencia, CA, USA), according to the manufacturer’s instructions. A total of 245 pooled swabs were subjected to viral RNA extraction, in which an individual pooled sample was prepared from approximately 4–5 swabs of the same specimen types. The influenza genome was examined by real-time RT-PCR using primer and probe sequences targeting the conserved region of influenza matrix protein, following the 2009 Centers for Disease Control and Prevention protocol (CDC protocol) [18]. RT-PCR amplification was performed in a QuantStudio 3 Real-Time PCR System using an AgPath-ID™ One-Step RT-PCR Kit (ThermoFisher Scientific, Waltham, MA, USA) with optimized quantitative RT-PCR mixtures (a 25 μL volume containing 12.5 μL of 2X master mix, 1 μL of 25X RT-PCR enzymes mix, 5 μL of extracted RNA, 0.8 μM each of forward and reverse primers, and 0.2 μM of the labeled probe and added RNase-free water to bring up the final volume). The amplification cycle was 50 °C for 30 min for reverse transcription and 95 °C for 10 min for Taq polymerase activation, followed by 45 cycles of PCR amplification (95 °C for 15 s and 55 °C for 30 s). Positive control (human H1 (104) genomic RNA) and negative control (nuclease-free water) were included in each run. The results were analyzed by QuantStudio™ Desing&Analysis Software (QuantStudio). The cycle threshold (Ct) value ≤ 40 was considered as a positive result.

### 2.7. Statistical Analyses

Microsoft Office Excel 2019 was used for data management and GraphPad Prism 8 was used for the GMT (95% CI) data analysis. The prevalence of IAV was calculated as the proportion of the positive results among the total number tested and demonstrated as a percentage. The Pearson chi-square test was used to compare the prevalence of seropositive IAV among animal age, animal sex, and subtype-specific antibodies. The one-way analysis of variance (ANOVA) was used to compare the level of HI antibody titer among subtypes of IAV. The difference with *p*-value < 0.05 was considered to be statistically significant.

## 3. Results

The HI and NT results demonstrated that 19.0% (128/672) of macaque serum samples had antibodies against the influenza A virus. Antibodies to a variety of influenza A virus subtypes were detected in the tested serum samples consisting of antibodies against AIV H1 (1.6%, 11/672), AIV H2 (15.2%, 102/672), AIV H3 (0.3%, 2/672), AIV H9 (3.4%, 23/672), and human H1 (NP-045) (0.9%, 6/672) (Table 2 and Appendix A). The statistically significant difference in the number of seropositive samples to AIV H2 subtype was observed (Pearson chi-square test, *p* < 0.01). Overall, seropositive samples showed the HI and NT titers in the range of 20–80 and 20–40, respectively. Exceptionally, one positive sample against AIV H9 showed a high titer of both HI and NT at 320 and 160, respectively (Figure 1 and Appendix A). The GMT (95% CI) of HI titers against each virus subtype were determined as AIV H1 = 10.13 (10.05–10.22), AIV H2 = 11.89 (11.50–12.30), AIV H3 = 10.03 (9.97–10.08), AIV H9 = 10.37 (10.19–10.55), and human H1 (NP-045) = 10.11 (10.02–10.21). Similarly, the GMT (95% CI) of NT titers against each virus subtype were also determined as AIV H1 = 22.69 (18.79–27.39), AIV H2 = 20 (20–20), AIV H3 = 20 (20–20), AIV H9 = 22.56 (18.57–27.42), and human H1 (NP-045) = 20 (20–20). A significant difference in HI antibody level was found in AIV H2 subtypes (ANOVA one-way test, *p* < 0.01). 

Among the 128 positive serum samples, 15 (11.7%) individual samples were found antibodies against more than one subtype of influenza A virus which AIV H1, H2, H3, H9, and human H1 (NP-045) were identified. Of those, 14 samples showed seropositivity against two subtypes representing 93.3% (14/15). Combinations of two subtypes found in the positive samples comprised AIV H1 + AIV H2 (40.0%, 6/15), AIV H1 + AIV H3 (6.7%, 1/15), AIV H2 + AIV H9 (26.7%, 4/15), and AIV H2 + human H1 (NP-045) (20.0%, 3/15). Interestingly, multiple serotypes against three subtypes identified as AIV H1 + AIV H2 + AIV H3 were observed in one positive sample representing 6.7% (1/15) (Table 2 and Appendix A). On the other hand, antibodies against the other subtypes of AIV including H4, H5, H6, H7, H8, H10, H11, H12, H13, H14, H15, H16, and human H1 (104) virus were not detected in all of the tested serum samples. 

Seropositive macaques were derived from 24 free-ranging colonies distributed in 13 provinces located in the north (*n* = 1), northeast (*n* = 3), east (*n* = 1), central (*n* = 4), west (*n* = 1), and south (*n* = 3) of Thailand (Table 2 and Figure 2). Conversely, macaques derived from the other three investigated colonies in the northeast (*n* = 1) and south (*n* = 2) provinces were free of IAV antibodies. The seropositivity rate observed among positive colonies ranged from 4–56%. The lowest seropositivity (4%, 1/25) was found in macaque colonies in Chon Buri and Petchaburi province located in the east and south of Thailand, respectively, whereas the highest seropositivity (56%, 14/25) was identified in another macaque colony in Chon Buri province. The prevalence of seropositive IAV was not different between sub-adult (20.2%, 47/233) and adult (18.9%, 67/355) animals (Pearson chi-square test, *p* = 0.697). On the contrary, the prevalence of IAV seropositive was significantly different between female (28.1%, 54/192) and male (15.6%, 66/422) animals (Pearson chi-square test, *p* = 0.01) (Table 2).

The highest number of seropositive sampled macaques was derived from AIV H2 subtype infection. The AIV H2-seropositive animals were found in 19 out of 24 positive colonies distributed in all regions of Thailand. The seropositive rate for AIV H2 was 4–48% among positive colonies. Of those, there were four positive colonies showing only antibodies against the AIV H2 subtype with a 16–32% seropositive rate. The four colonies were all settled in temples of Lop Buri (*n* = 3) and Phatthalung (*n* = 1) provinces located in the central and south of Thailand, respectively. Notably, as many as four different seropositive subtypes could be found in an individual colony in Chon Buri and Si Sa Ket provinces. Seropositivity to four AIV subtypes (H1, H2, H3, and H9) were identified in a Chon Buri colony. The combination of two subtypes (H1 + H2, H1 + H3, and H2 + H9) and three subtypes (H1 + H2 + H3) was carried by macaques in this colony. Additionally, an antibody to the AIV H3 subtype was found only in the Chon Buri colony. Furthermore, macaques from a Si Sa Ket colony showed seropositivity to three AIV subtypes (H1, H2, and H9) and one human H1 subtype (NP-045). The combination of two subtypes (H1 + H2 and H2 + NP-045) was observed in the positive animals. The location of these two colonies was in the community area and close to the local district offices (Appendix A). 

Our genomic investigation demonstrated that in the 245 pooled swab samples, one (0.41%) from the pooled rectal swab samples was positive for the influenza virus with a cycle threshold (Ct) value of 38 (limit of detection was 45). The genomic RNA was subsequently extracted from an individual rectal swab sample of a positive pooled sample. We observed the positive real-time RT-PCR result with a Ct value of 38 in a rectal swab sample obtained from a sub-adult male macaque living in Kosumphi Forest Park, Maha Sarakham located in the northeastern part of Thailand. However, attempts to amplify longer target genes (matrix and hemagglutinin genes) or virus isolation from the positive rectal swab sample were unsuccessful. Moreover, no antibodies against the subtypes of IAV tested were found in this positive macaque. 

## 4. Discussion

Information on IAV infection in NHP populations is still limited, particularly in the wild populations of this animal species. Evidence for IAV exposure has been previously demonstrated through antibody detection in either macaques or chimpanzees in several countries, including Bangladesh, Singapore, Indonesia, and the Netherlands [10,11]. Antibodies against avian influenza virus (H9N2) and human influenza viruses (H1N1 and H3N2) have been identified in seropositive animals [10,11]. Seropositivity reported in those studies has been defined based on the HI assay with the cut-off titer of ≥1:10. In our current study, we used HI assay as the screening test with the higher cut-off titer of 1:20 to detect antibodies against 16 subtypes of AIV and two strains of human H1N1 influenza virus. The HI-positive samples with HI titer of ≥1:20 were further determined for neutralizing (NT) antibodies using an MN assay. The sample with both HI and NT titers of ≥20 was considered positive. Our results showed that the GMT of HI titers in seropositive samples ranged from 10.03–11.89; while the GMT of HI titers in seronegative samples were equal to 10. On the other hand, the GMT of NT titers in seropositive samples were ranged from 20–22.69. To our knowledge, this is the first report to determine the level of NT antibodies in wild NHP populations. 

In addition, the IAV exposure in NHPs has also been addressed through the molecular detection of the influenza virus [10]. The influenza virus genome has been successfully identified in a buccal swab from an adult macaque in Cambodia [10]. Conversely, our study could detect the influenza virus genome in a rectal swab sample obtained from a sub-adult male macaque living in Kosumphi Forest Park, Maha Sarakham. However, we could not amplify longer target genes (matrix and hemagglutinin genes) or virus isolation in the MDCK cell and embryonated eggs from the positive swab sample. Although the presence of IAV in rectal swabs is rare, several studies have reported the detection of influenza A viral RNA in either rectal swab samples or fecal samples obtained from humans [19,20,21,22]. However, no previous influenza A viral RNA has been observed in rectal swabs from NHPs. 

The prevalence of antibodies to AIV H1 and H3 and human H1 (NP-045) among the cynomolgus macaque populations in this study was consistent with the evidence in domestic avian species and the human population in Thailand [23,24,25]. The presence of seropositivity to AIV H2 and AIV H9 in free-ranging macaques could raise public concerns. The H2 and H9 influenza virus could be mutated and reassorted with other circulating subtypes promoting the emergence of strains with pandemic potential [26,27]. Thus, an active influenza surveillance program to identify reservoir hosts of AIV H2 and AIV H9 virus is needed. It is unclear why cynomolgus macaques are more susceptible to being infected with AIV H2 than the other subtypes. Most habitats of seropositive free-ranging macaques are close to temples, tourist places, and community areas (Appendix A). The source of IAV infection in cynomolgus macaques is unknown, but most likely came from avian species and human populations, which share the same habitats/environment. A recent macaque population survey has reported that about 50,000 individuals in Thailand live in the area of human–macaque conflict [14]. The conflict commonly arises when humans and wild animals are forced to share space and resources. The adverse consequences of conflict impacted all relevant sectors, including humans, macaques, and the environment through habitat disturbance and destruction, crop and property damages, and potential threats to health and well-being [14]. There are currently over 200 human–macaque conflict areas in Thailand, almost 80% of which have already been surveyed for macaque populations [14]. The close proximity between macaque colonies and the human community could pose a potential risk for virus transmission between macaques and humans. 

Defining the ecology of IAV infection in NHP populations is crucial. The continued active surveillance will provide information on virus exposure and extend the knowledge of virus ecology in the target populations. Although our previous study has demonstrated no evidence of IAV exposure among NHP populations in Thailand [12], we still considered continuing the investigation particularly in free-ranging colonies which are close to human communities. In our present study, we attempted to expand the sampling population and antibody detection against more virus subtypes in order to overcome the limitation in our previous study and to support the information on the virus ecology and epidemiology in the target populations. 

However, our study had some limitations. Firstly, the samples obtained from the current study may not represent the entire NHP population in Thailand. The results should be carefully interpreted concerning the sample size issue. Secondly, there was a lack of local avian influenza virus strains used in the serological tests. At present, various LPAI subtypes, including H1, H3, H4, H7, H8, H10, and H11, have been identified in domestic avian species [23,24,28,29,30,31] and the LPAI subtype of H12 has been successfully isolated from a watercock and lesser whistling duck in Thailand [32]. Based on our virus resources, we used 16 influenza HA subtypes (H1–H16) of the AIV and two Thailand strains of the human H1N1 influenza virus as the tested antigen. The serological results obtained from the study might be inadequate to reflect the current situation for the virus in the target populations. Nevertheless, our findings clearly revealed the role of cynomolgus macaques in virus ecology. Further, serological investigation using the recent local strains of influenza viruses will help to update the current virus circulation in the NHP populations. This information may help facilitate a national strategic plan for disease control as well as NHP population control in the future.

## 5. Conclusions 

Our findings indicated evidence for the natural infection of avian and human influenza viruses in cynomolgus macaques in Thailand. The serological investigation suggested that cynomolgus macaques were susceptible to multiple influenza virus subtypes, including AIV H1, H2, H3, H9, and human H1 (NP-045); some of these infected subtypes presented here have not been previously reported in this animal species. An influenza genome was found in rectal swab samples, indicating active influenza infection at the time of sample collection. Our results highlight the role of cynomolgus macaques in the ecology of IAV and the need for continuous surveillance of IAV in NHP populations, particularly in the animal colonies where their habitat is shared with humans and domestic avian species. 

## Figures and Tables

**Figure 1 vetsci-09-00132-f001:**
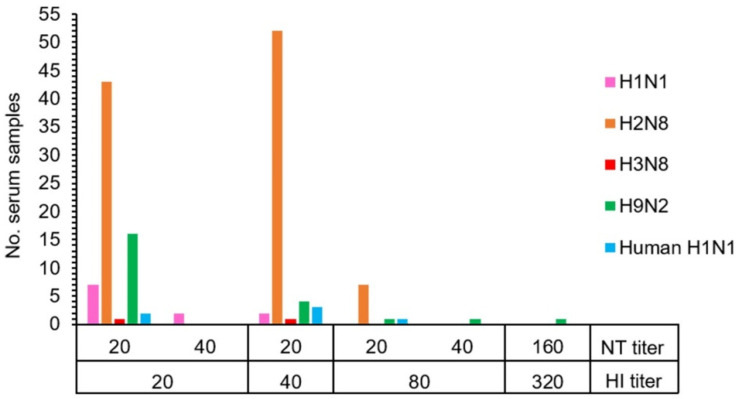
The correlation between hemagglutination inhibition (HI) and neutralization (NT) antibody titers of 128 individual macaques with seropositivity against subtypes of influenza A virus.

**Figure 2 vetsci-09-00132-f002:**
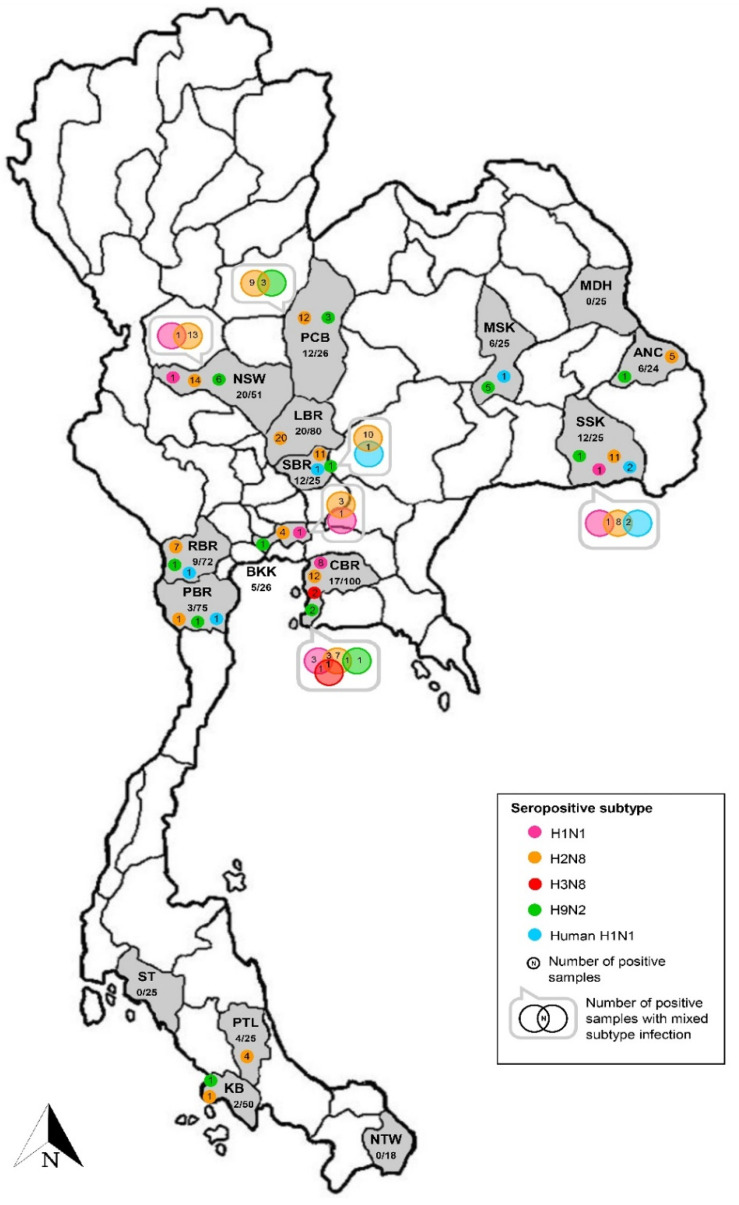
Seropositivity against influenza virus subtypes found in the investigated macaque serum samples. Provinces where the habitat of macaques are shown as abbreviations, including ANC: Amnat Charoen, BKK: Bangkok, CBR: Chon Buri, KB: Krabi, LBR: Lop Buri, MDH: Mukdahan, MSK: Maha Sarakham, NSW: Nakhon Sawan, NTW: Narathiwat, PBR: Phetchaburi, PCB: Phetchabun, PTL: Phatthalung, RBR: Ratchaburi, SBR: Saraburi, SSK: Si Sa Ket, and ST: Satun. In each province, the number of positive samples/total number of tested samples and number of positive samples against influenza virus subtypes are indicated.

**Table 1 vetsci-09-00132-t001:** Demographic characteristics of sera derived from free-ranging cynomolgus macaques, Thailand, March–November 2019.

Habitat Locations		Sample No.	Age *			Sex		
Province	Place (No. of Investigated Colonies)		Sub-Adult	Adult	NA	Female	Male	NA
Amnat Charoen	Botanical park (1)	24	-	-	24	16	8	-
Bangkok	Community area (1)	26	4	22	-	7	19	-
Chon Buri	Temple (2), Community area (2)	100	47	53	-	22	78	-
Krabi	Tourist place (2)	50	25	25	-	15	35	-
Lop Buri	Temple (3)	80	20	60	-	35	45	-
Maha Sarakham	Forest park (1)	25	13	12	-	4	21	-
Mukdahan	Temple (1)	25	-	-	25	-	-	25
Nakhon Sawan	Temple (1), Tourist place (1)	51	17	34	-	11	40	-
Narathiwat	Tourist place (1)	18	6	12	-	10	8	-
Phatthalung	Temple (1)	25	14	11	-	9	16	-
Phetchabun	Temple (1)	26	12	14	-	6	20	-
Phetchaburi	Temple (2), Tourist place (1)	75	29	44	2	15	60	-
Ratchaburi	Community area (1),	72	22	25	25	11	36	25
	Tourist place (1), Stone Park (1)							
Saraburi	Temple (1)	25	10	15	-	8	17	-
Satun	Park (1)	25	14	6	5	3	17	5
Si Sa Ket	Community area (1)	25	0	22	3	20	2	3
Total		672	233	355	84	192	422	58

*: Sub-adult was defined as macaque with the age of 1–4 years, while the adult was defined as macaque with age > 4 years; NA: not available.

**Table 2 vetsci-09-00132-t002:** Seropositivity against influenza A virus subtypes among free-ranging cynomolgus macaques, Thailand, March–November 2019.

Virus Strain	Virus Subtype	No. of Seropositive *			Age **		Sex ***			
		Overall (%)	Single Subtypes	Mixed Subtypes	Sub-adult	Adult	NA	F	M	NA
A/Aquatic bird/Hong Kong/DI25/2002	H1N1	11 (1.6%)	3	8	5	6	-	5	6	-
A/Wild Duck/Shan Tou/992/2000	H2N8	102 (15.2%)	88	14	36	56	10	48	49	5
A/Duck/Shan Tou/1283/2001	H3N8	2 (0.3%)	0	2	2	-	-	1	1	-
A/Duck/Shan Tou/461/2000	H4N9	0 (0%)	-	-	-	-	-	-	-	-
A/Duck/Jiangxi/6151/2003	H5N3	0 (0%)	-	-	-	-	-	-	-	-
A/Heron/Hong Kong/LC10/2002	H6N8	0 (0%)	-	-	-	-	-	-	-	-
A/Ostrich/Zimbabwe/222/1996	H7N1	0 (0%)	-	-	-	-	-	-	-	-
A/Mallard/Alberta/242/2003	H8N4	0 (0%)	-	-	-	-	-	-	-	-
A/Chicken/Hong Kong/G9/1997	H9N2	23 (3.4%)	19	4	8	12	3	7	14	2
A/Duck/Shan Tou/1796/2001	H10N8	0 (0%)	-	-	-	-	-	-	-	-
A/Duck/Shan Tou/1411/2000	H11N2	0 (0%)	-	-	-	-	-	-	-	-
A/Red-necked stint/Australia/5745/1981	H12N9	0 (0%)	-	-	-	-	-	-	-	-
A/Gull/MD/704/1977	H13N6	0 (0%)	-	-	-	-	-	-	-	-
A/Mallard/Gurjev/263/1982	H14N5	0 (0%)	-	-	-	-	-	-	-	-
A/Duck/Australia/341/1983	H15N8	0 (0%)	-	-	-	-	-	-	-	-
A/Shorebird/DE/172/2006	H16N3	0 (0%)	-	-	-	-	-	-	-	-
A/Thailand/104/2009	H1N1	0 (0%)	-	-	-	-	-	-	-	-
A/Thailand/NP-045/2018	H1N1	6 (0.9%)	3	3	1	4	1	2	3	1
Total (by individual sample)		144 (21.4%)	113	31	52	78	14	63	73	8
(by individual animal)		128 (19.0%)	113	15	47	67	14	54	66	8

*: Seropositive was defined as both HI and NT antibody titers of ≥20; multiple serotypes in individual samples including H1 + H2 (*n* = 6), H1 + H3 (*n* = 1), H2 + H9 (*n* = 4), H2 + Human H1 (NP-045) (*n* = 3), H1 + H2 + H3 (*n* = 1). **: Sub-adult was defined as macaque with the age of 1–4 years, while the adult was defined as macaque with age > 4 years; NA: not available. ***: F: Female; M: Male; NA: not available.

## Data Availability

The data presented in the study are available in the manuscript.

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
