# Peer review of "Evidence of Influenza A Virus Infection in Cynomolgus Macaques, Thailand"

_vetsci, 2022, doi:10.3390/vetsci9030132_

Round 1

Reviewer 1 Report

While most of the world is focused on the COVID-19 pandemic and SARS-CoV-2 spread, it should be highlighted that there are many other pathogens with pandemic potential, such as influenza viruses. Identification of IAV animal reservoirs are of high interest for the potential pandemic control. Paungpin and colleagues demonstrated that the free ranging non-human primates in Thailand can be infected with influenza viruses and play a vital role in the IAV circulation in the environment. The manuscript is well written and the results are clear. Below you can find my minor comments:

  1. Table 1 – the numbers in the brackets are unclear. Why i.e. the NHP colonies in Chon Buri has number ‘2’ and ‘2’? Similar situation is with colonies at Nakhon Sawan and Ratchaburi.
  2. Please describe the statistical analysis in the Material and Methods section.
  3. The numbers of positive samples on Figure 2 in some are unclear. I’m not sure whether it’s my setting or authors should use better quality picture.
  4. And last but not least – I wonder why authors decided to use the IAV isolates mostly from China (Table 2)?

Author Response

Response to reviewers

 We would like to thank the reviewers for their comments for the manuscript. We agree that the comments have identified important point which required an improvement. After completion of the suggested edits, the revised manuscript will be greatly improved.

We have responded below in detail to each reviewers’ comments including how and where the text was modified using RED color. The changes in the revised manuscript were addressed by using track change as journal’s suggestion.  

We hope that the reviewers will find our responses to their comments satisfactory, and we are willing to finish the revised version of the manuscript including any further suggestions that the reviewers may have.

Thank you very much for reviewing our manuscript and we are looking forward to hearing from you soon.

Yours sincerely,

Witthawat Wiriyarat

(corresponding author)

Reviewer 1

While most of the world is focused on the COVID-19 pandemic and SARS-CoV-2 spread, it should be highlighted that there are many other pathogens with pandemic potential, such as influenza viruses. Identification of IAV animal reservoirs are of high interest for the potential pandemic control. Paungpin and colleagues demonstrated that the free ranging non-human primates in Thailand can be infected with influenza viruses and play a vital role in the IAV circulation in the environment. The manuscript is well written and the results are clear. Below you can find my minor comments:

Comment 1: Table 1 – the numbers in the brackets are unclear. Why i.e. the NHP colonies in Chon Buri has number ‘2’ and ‘2’? Similar situation is with colonies at Nakhon Sawan and Ratchaburi.

Response: According to your comment, we would like to clarify the number in the brackets presented in Table 1. The numbers in the brackets are represented the number of macaque colonies in an individual habitat where the animals were investigated. For example, Chon Buri has 4 investigated macaque colonies including 2 colonies at temples and 2 colonies at community areas. Therefore, we have described the number of macaque colonies in each habitat location in Chon Buri as Temple (2) and Community area (2) as shown in Table 1. On the other hand, Nakhon Sawan has 2 investigated macaque colonies including 1 colony at temple and 1 colony at tourist place. The number of macaque colonies in each habitat location in Nakhon Sawan were represented as Temple (1) and Tourist place (1). In case of Ratchaburi, we investigated 3 macaque colonies in this province which the number of colonies in each habitat locations were defined as Community area (1), Tourist place (1) and Stone park (1). Additionally, we have provided more detail of individual habitat location in each province in supplementary Table S1.

However, to clarify the meaning of the number in the brackets presented in Table 1, we have corrected the table header “Place (Colony No.)” with “Place (No. of investigated colonies)”.

Comment 2: Please describe the statistical analysis in the Material and Methods section.

Response: According to your suggestion, we have described the statistical analysis in the Material and Methods section, Line 153-160 as follows:

2.7 Statistical analyses

Microsoft Office Excel 2019 was used for data management and GraphPad Prism 8 was used for the GMT (95% CI) data analysis. The prevalence of IAV was calculated as the proportion of the positive results among the total number tested and demonstrated as a percentage. The Pearson Chi-Square test was used to compare the prevalence of IAV seropositive among animal age, animal sex, and subtype-specific antibodies. The one way ANOVA was used to compare the level of HI antibody titer among subtypes of IAV. The difference with p-value < 0.05 was considered to be statistically significant.   

Comment 3: The numbers of positive samples on Figure 2 in some are unclear. I’m not sure whether it’s my setting or authors should use better quality picture.

Response: According to your comment, we have improved the resolution of Figure 2 which the numbers of positive samples are clearer to read.

Comment 4: I wonder why authors decided to use the IAV isolates mostly from China (Table 2)?

Response: We thank the reviewer for the comment. Since we lacked the local isolates of avian influenza viruses (AIV), we decided to use the 16 subtypes of AIV kindly provided by Professor Robert G. Webster, St. Jude Children Research Laboratory, Tennessee, USA through our co-investigator, Dr. Pilaipan Puthavathana. In addition, we used 2 local isolates of human H1N1 virus. Based on the available virus resources, we strongly agree with the reviewer that the IAV isolates used in the study were mostly from China. However, the primary aim of the present study was to determine the natural exposure of IAV in cynomolgus macaques population. In this case, HI assay was performed to identify the hemagglutinin (HA) subtype specific antibodies to influenza virus. The tested viruses that we included in the HI assay could serve to represent 16 HA subtypes of AIV and H1 subtype of human influenza A virus. Nevertheless, we have considered that the investigation using local isolates or recent isolates originated worldwide could strengthen the surveillance program for monitoring the virus circulation as well as reflecting the current situation of virus infection in the targeted population.

Reviewer 2 Report

Reviewer comments on the manuscript vetsci-1604328 entitled: Evidence of influenza A virus infection in nonhuman primates (NHPs), Thailand
submitted by Witthawat Wiriyarat and colleagues for consideration in Veterinary Sciences. 
This manuscript describes an active surveillance in non-human primates (NHP) in 27 free-ranging colonies in 16 provinces in Thailand. 
627 serum samples were collected from March to November 2019. Samples were tested by HI test and further confirmed by MN assay against a panel of AIV H1 to H16. About 21% of samples from 13/16 provinces were tested seropositive with high prevalence for H2 and H9 AIV and human H1N1.
Moreover, swab samples from 245 pools were tested for AIV M-gene using RT-qPCR. Low viral RNA load was detecetd in one sample only.

The manuscript is well written and provides valuable information on a potential IAV reservoir or mixing-vessel host, which in close contact to humans. It includes important information on the ecology of NHP in Thailand.

The following comments should be addressed:

1. "LPAI" should be changed to "AIV". HPAIV is only for H5/H7 subtypes. Secondly, serologically, it is not possible to defferntiate between LP or HP antibodies.
2. Line 123: "HI titers ≥5120 were assigned as 5120". There is no similar titer in the manuscript. Please rephrase
3. Supplementary Table S1 is not useful because all data are already described in Table 2. Please remove and change the text accordingly or if add more information in Tables S1 (Accession numbers, source of the virus, etc.)
4. Detection of IAV in rectal swabs is very rare. In the discussion section, it would be very helpful to cite a reference for the detection of IAV RNA in rectal swabs in humans or NHP.
5. Line 268: Please explain "human-macaques conflict"?
6. The viral RNA was detected in one animal. It will be helpful to mention if this animal had antibodies or not. If it was seropositive, please mention against which subtype.

Author Response

We would like to thank the reviewers for their comments for the manuscript. We agree that the comments have identified important point which required an improvement. After completion of the suggested edits, the revised manuscript will be greatly improved.

We have responded below in detail to each reviewers’ comments including how and where the text was modified using RED color. The changes in the revised manuscript were addressed by using track change as journal’s suggestion.  

We hope that the reviewers will find our responses to their comments satisfactory, and we are willing to finish the revised version of the manuscript including any further suggestions that the reviewers may have.

Thank you very much for reviewing our manuscript and we are looking forward to hearing from you soon.

Yours sincerely,

Witthawat Wiriyarat

(corresponding author)

Reviewer 2

Reviewer comments on the manuscript vetsci-1604328 entitled: Evidence of influenza A virus infection in nonhuman primates (NHPs), Thailand submitted by Witthawat Wiriyarat and colleagues for consideration in Veterinary Sciences. This manuscript describes an active surveillance in non-human primates (NHP) in 27 free-ranging colonies in 16 provinces in Thailand.  627 serum samples were collected from March to November 2019. Samples were tested by HI test and further confirmed by MN assay against a panel of AIV H1 to H16. About 21% of samples from 13/16 provinces were tested seropositive with high prevalence for H2 and H9 AIV and human H1N1.

Moreover, swab samples from 245 pools were tested for AIV M-gene using RT-qPCR. Low viral RNA load was detected in one sample only.

The manuscript is well written and provides valuable information on a potential IAV reservoir or mixing-vessel host, which in close contact to humans. It includes important information on the ecology of NHP in Thailand.

The following comments should be addressed:

Comment 1: "LPAI" should be changed to "AIV". HPAIV is only for H5/H7 subtypes. Secondly, serologically, it is not possible to differentiate between LP or HP antibodies.

Response: Thank you very much for your constructive comment. According to your comment, we have changed the “LPAI” to “AIV” throughout the manuscript.

Comment 2: Line 123: "HI titers ≥5120 were assigned as 5120". There is no similar titer in the manuscript. Please rephrase.

Response: We thank the reviewer for the comment. According to your comment, we have revised the sentence in the Line 122 as below:

The original sentence:

The HI titers < 20 were assigned as 10, and the HI titers ≥5120 were assigned as 5120.

The revised sentence:

The HI titers < 20 were assigned as 10, and the HI titers ≥320 were assigned as 320.

Comment 3: Supplementary Table S1 is not useful because all data are already described in Table 2. Please remove and change the text accordingly or if add more information in Tables S1 (Accession numbers, source of the virus, etc.)

Response: According to your comment, we have removed supplementary Table S1 and changed the text accordingly throughout the manuscript.

Comment 4: Detection of IAV in rectal swabs is very rare. In the discussion section, it would be very helpful to cite a reference for the detection of IAV RNA in rectal swabs in humans or NHP.

Response: Thank you very much for your constructive comment. According to your comment, we have addressed and cited the references related to the detection of IAV RNA in rectal swabs in humans or NHP in the discussion section, Line 271-275 as follows:

Although the presence of IAV in rectal swabs is very rare, several studies have reported the detection of influenza A viral RNA in either rectal swab samples or fecal samples obtained from humans [19-22]. However, no influenza A viral RNA has been observed earlier in rectal swabs from NHPs.

According to the additional discussion we mentioned above, the new references corresponding to the discussion have been added in the section of References including Reference No. 19-22.

Comment 5: Line 268: Please explain "human-macaques conflict"?

Response: According to your comment, we have added more explanation on the “human-macaques conflict” in the discussion section, Line 289-293 as follows:

The conflict commonly arises when human and wild animals are forced to share space and resources. The adverse consequences of conflict impacted all relevant sectors including humans, macaques and environment through habitat disturbance and destruction, crop and property damages and potential threats to health and well-being [14]. 

Comment 6: The viral RNA was detected in one animal. It will be helpful to mention if this animal had antibodies or not. If it was seropositive, please mention against which subtype.

Response: Thank you very much for your constructive comment. According to your comment, we have mentioned on the result of antibody detection in the RT-PCR positive animal in the result section, Line 238-239 as follows:

Besides, no antibodies against the subtypes of IAV tested were found in this positive macaque.

Reviewer 3 Report

Comments

The “Evidence of influenza A virus infection in nonhuman primates (NHPs), Thailand” manuscript is a well-written and informative study showing the surveillance of IAV among 672 cynomolgus macaques living in 27 free-ranging colonies in Thailand between March and November 2019, Using the hemagglutination inhibition and the serum neutralization tests for seroprevalence and the reverse transcriptase quantitative polymerase chain reaction RT-PCR for molecular detection of the matrix protein gene of the influenza A virus. The manuscript is providing evidence of cynomolgus macaques infection with H1, H2, H3, and H9 influenza viruses in Thailand through seroprevalence testing.

I am recommending the publication of this manuscript with minor revisions:

  1. The title;

Since the surveillance was done using the cynomolgus macaques species only, I am recommending the change of the title to Evidence of influenza A virus infection in cynomolgus macaques, Thailand.

Accordingly, the NHPs in the introduction (line 70) need to be replaced by cynomolgus macaques and throughout the manuscript.

  1. The introduction;

The authors mentioned a yearlong investigation of IAV exposure (Lines 69 and 70) and in the material and methods (lines 77&78) the samples collection was done between March and November of 2019. The time in the introduction needs to be consistent with the time in the material and methods.

  1. The material and methods:

Although the use of the HI and SN tests for the detection of the seroprevalence is valid, these tests are strain-specific and will provide the prevalence of the used strains only. The use of influenza A nucleoprotein ELISA assay can provide more comprehensive evidence for the exposure of the macaques to any IAV. I am suggesting presenting the sera ELISA results if it is available or providing a justification for not using this assay.

Author Response

We would like to thank the reviewers for their comments for the manuscript. We agree that the comments have identified important point which required an improvement. After completion of the suggested edits, the revised manuscript will be greatly improved.

We have responded below in detail to each reviewers’ comments including how and where the text was modified using RED color. The changes in the revised manuscript were addressed by using track change as journal’s suggestion.  

We hope that the reviewers will find our responses to their comments satisfactory, and we are willing to finish the revised version of the manuscript including any further suggestions that the reviewers may have.

Thank you very much for reviewing our manuscript and we are looking forward to hearing from you soon.

Yours sincerely,

Witthawat Wiriyarat

(corresponding author)

Reviewer 3

The “Evidence of influenza A virus infection in nonhuman primates (NHPs), Thailand” manuscript is a well-written and informative study showing the surveillance of IAV among 672 cynomolgus macaques living in 27 free-ranging colonies in Thailand between March and November 2019, Using the hemagglutination inhibition and the serum neutralization tests for seroprevalence and the reverse transcriptase quantitative polymerase chain reaction RT-PCR for molecular detection of the matrix protein gene of the influenza A virus. The manuscript is providing evidence of cynomolgus macaques infection with H1, H2, H3, and H9 influenza viruses in Thailand through seroprevalence testing.

I am recommending the publication of this manuscript with minor revisions:

Comment 1: The title;

Since the surveillance was done using the cynomolgus macaques species only, I am recommending the change of the title to Evidence of influenza A virus infection in cynomolgus macaques, Thailand.

Accordingly, the NHPs in the introduction (line 70) need to be replaced by cynomolgus macaques and throughout the manuscript.

Response: Thank you very much for your constructive comment. We agree with your suggestion to change the title of the manuscript to “Evidence of influenza A virus infection in cynomolgus macaques, Thailand”.

Accordingly, we have replaced the term “NHPs” with “cynomolgus macaques” in the introduction (line 70) and also other relevant parts throughout the manuscript. However, we still used the term “NHPs” in some sentences when we mentioned in general contexts about this animal species.

In addition, we have added “cynomolgus macaques” in Keywords, Line 29 of the manuscript.

Comment 2: The introduction;

The authors mentioned a yearlong investigation of IAV exposure (Lines 69 and 70) and in the material and methods (lines 77&78) the samples collection was done between March and November of 2019. The time in the introduction needs to be consistent with the time in the material and methods.

Response: We thank the reviewer for the comment. According to your comment, we have revised the sentence in the introduction section, Line 68-69 as below:

The original sentence:

Thus, we performed an intensive year-long investigation of IAV exposure using serological and molecular methods in free-ranging NHP populations in Thailand.

The revised sentence:

Thus, we performed an investigation of IAV exposure using serological and molecular methods in free-ranging cynomolgus macaques in Thailand between March and November 2019.

 Comment 3: The material and methods:

Although the use of the HI and SN tests for the detection of the seroprevalence is valid, these tests are strain-specific and will provide the prevalence of the used strains only. The use of influenza A nucleoprotein ELISA assay can provide more comprehensive evidence for the exposure of the macaques to any IAV. I am suggesting presenting the sera ELISA results if it is available or providing a justification for not using this assay.

Response: Thank you very much for your constructive comment. We strongly agree with the reviewer that the use of influenza A nucleoprotein ELISA assay can provide more comprehensive evidence for the exposure of the macaques to any IAV since the nucleoprotein of IAV is conserved protein. The result obtained from the ELISA assay gives information about the binding antibodies. However, our study aimed to demonstrate serosubtype-specific antibodies found in macaques by using HI and NT assays. The results obtained from the HI and NT assays can determine the level of protective antibodies which show neutralizing activity in the investigated animals.

Moreover, the use of the ELISA test kit as screening tool for serological surveillance of influenza virus should be evaluated in particular when the tested sera were derived from multi animal species. In this case, we have experienced to use a commercially species-independent test kit as the screening tests for our previous study of influenza virus exposure in felid [1].  In the study, we found that 5 sera which were HI positive but ELISA negative showed NT antibody titers in the range of 40–160 [1]. Thus, those 5 sera with negative screening by ELISA should be considered as truly positive. 

Discrepancies between the results of indirect ELISA and surrogate virus neutralization test (sVNT) have been also demonstrated in other pathogens such as SARS-CoV-2 in animal [2-4]. However, other factors such as background of the animals, the timing of animal infection and serum sample collection may affect the results of ELISA and VNT test. According to the above reasons, we decided not to use the ELISA assay in our current study.   

Referrences

  1. Sangkachai, N., Thongdee, M., Chaiwattanarungruengpaisan, S., Buddhirongawatr, R., Chamsai, T., Poltep, K., Wiriyarat, W. and Paungpin W. (2019) Serological evidence of influenza virus infection in captive wild felids, Thailand. J. Vet. Med. Sci., 81(9): 1341-1347.
  2. Michelitsch, A., Hoffmann, D., Wernike, K., & Beer, M. (2020). Occurrence of antibodies against SARS-CoV-2 in the domestic cat population of Germany. Vaccines (Basel), 8(4). https://doi.org/10.3390/vaccines8040772.
  3. Zhang, Q., Zhang, H., Gao, J., Huang, K., Yang, Y., Hui, X., He, X., Li, C., Gong, W., Zhang, Y., Zhao, Y., Peng, C., Gao, X., Chen, H., Zou, Z., Shi, Z.-L., & Jin, M. (2020). A serological survey of SARS-CoV-2 in cat in Wuhan. Emerging Microbes & Infection, 9(1), 2013–2019. https://doi.org/10.1080/ 22221751.2020.1817796.
  4. Zhao, Y., Yang, Y., Gao, J., Huang, K., Hu, C., Hui, X., He, X., Li, C., Gong, W., Lv, C., Zhang, Y., Chen, H., Zou, Z., Zhang, Q., & Jin, M. (2021). A serological survey of severe acute respiratory syndrome coronavirus 2 in dogs in Wuhan. Transboundary and Emerging Diseases, https://doi.org/10.1111/ tbed.14024.

Round 2

Reviewer 2 Report

The authors answered all my comments satisfactorily